# Fasudil Increased the Sensitivity to Gefitinib in NSCLC by Decreasing Intracellular Lipid Accumulation

**DOI:** 10.3390/cancers14194709

**Published:** 2022-09-27

**Authors:** Tingting Liao, Jingjing Deng, Wenjuan Chen, Juanjuan Xu, Guanghai Yang, Mei Zhou, Zhilei Lv, Sufei Wang, Siwei Song, Xueyun Tan, Zhengrong Yin, Yumei Li, Yang Jin

**Affiliations:** 1NHC Key Laboratory of Pulmonary Diseases, Key Laboratory of Biological Targeted Therapy, Ministry of Education, Department of Respiratory and Critical Care Medicine, Union Hospital, Tongji Medical College, Huazhong University of Science and Technology, 1277 Jiefang Avenue, Wuhan 430022, China; 2Department of Endocrinology, Union Hospital, Tongji Medical College, Huazhong University of Science and Technology, Wuhan 430022, China; 3Department of Thoracic Surgery, Union Hospital, Tongji Medical College, Huazhong University of Science and Technology, 1277 Jiefang Avenue, Wuhan 430022, China

**Keywords:** Fasudil, EGFR, NSCLC, gefitinib, lipid metabolism

## Abstract

**Simple Summary:**

The anti-tumor role of Fasudil in EGFR-mutation NSCLC as well as its mechanism are largely unknown. Here, we show that Fasudil could effectively inhibit EGFR-mutation cell growth and enhance the sensitivity of gefitinib-resistant NSCLC cells to gefitinib by suppressing intracellular lipid accumulation. Mechanistic investigations showed that Fasudil could reverse gefitinib-induced SCD1 expression by suppressing AMPK activity, and combination therapy had a greater inactivation effect on the EGFR/PI3K/AKT pathway than either treatment alone. These findings highlight the clinical significance of Fasudil in EGFR mutation NSCLC therapy.

**Abstract:**

Tyrosine kinase inhibitors (TKIs) resistance is a challenge in patients with epidermal growth factor receptor (EGFR)-mutant non-small-cell lung cancer (NSCLC). Here, we examined the effect of Fasudil in reversing TKIs resistance. The results of CCK8 assay, clone formation assay, cell cycle arrest analysis, and apoptosis analysis show that Fasudil treatment effectively suppressed the growth and induced apoptosis of the EGFR-mutant NSCLC cells. Furthermore, Fasudil in combination with gefitinib showed a synergistic anti-tumor effect in gefitinib-resistant NSCLC cells. RNA-seq analysis and immunoblotting indicated that Fasudil treatment significantly inhibited intracellular lipid accumulation and EGFR/PI3K/AKT pathway activation. Mechanistic investigations showed that Fasudil regulated lipogenic gene expressions via AMPK signal pathway. In vivo, Fasudil and gefitinib co-administration significantly attenuated the growth of H1975 nude mouse xenograft models, suggesting that Fasudil treatment combined with gefitinib can be applied as a therapy for gefitinib-resistant NSCLC cells.

## 1. Background

According to tumor data reported in 2020, lung cancer remains the leading cause of cancer-related deaths worldwide [1]. At present, the main treatment methods for lung cancer include surgical resection, chemotherapy, targeted therapy, immunotherapy, and supportive therapy, and for advanced lung cancer, traditional cytotoxic chemotherapy remains the most important treatment option. However, it is expected that more patients with lung cancer will benefit from targeted therapy, considering the extensive studies and recent developments that have been attained in this regard [2]. Potentially actionable target molecules have been identified in 69% of patients with advanced non-small-cell lung cancer (NSCLC), and mutations in the epidermal growth factor receptor (EGFR) gene are the most common, followed by KRAS, ALK (Anaplastic lymphoma kinase), MET (mesenchymal–epithelial transition), HER2, ROS1 (ROS proto-oncogene 1), BRAF (V-raf murine sarcoma viral oncogene homolog B), RET (Rearranged during Transfection), etc. [3].

The EGFR pathway regulates various cellular functions, including cancer cell proliferation, survival, metastasis, and angiogenesis, through the activation of its multiple downstream pathways, such as RAS/RAF-MEK (mitogen-activated protein kinase)-ERK1/2 (extracellular signal-regulated kinase), PI3K (phosphatidylinositol-4, 5-bisphosphate 3-kinase)/AKT (a serine/threonine protein kinase), and JAK2- STAT3/5 (signal transducer and activator of transcription) [4,5]. EGFR activation mutations mainly occur in exons 18, 19, 20, and 21, of which the deletion of exon 19 and exon 21 L858R point mutations are the most common EGFR activation mutations, accounting for 90% of all EGFR activation mutations. The frequency of EGFR activation mutations may vary from population to population, ranging from around 10% to 15% in Caucasian NSCLC populations [6]; in Asian NSCLC populations, the frequency of EGFR activation mutations is around 30% to 50%. EGFR activation mutations are most common in non-smoking, female Asian NSCLC patients [7].

EGFR Tyrosine Kinase Inhibitors (EGFR-TKIs) are first-line therapy for patients with EGFR-activating-mutations NSCLC, and they include gefitinib, erlotinib, afatinib, dacomitinib, and osimertinib. Gefitinib is a first-generation EGFR-TKI, which has good efficacy and tolerance for lung adenocarcinoma patients with EGFR-sensitive mutations and has been included in the Chinese medical insurance directory [8]. However, EGFR-TKI resistance is the greatest challenge for NSCLC patients with EGFR mutation in choosing TKI therapy, with approximately 10% of these patients showing primary TKI resistance, while 70% of them eventually develop acquired resistance to EGFR–TKIs [9]. To date, several underlying TKI resistance development mechanisms have been identified, among which the EGFR T790M mutation is the most common mechanism for the development of acquired resistance in the first and second generations of EGFR TKIs, present in 50% to 60% of drug-resistant cases. The T790 residue is located in the ATP (adenosine triphosphate) binding pocket of the EGFR protein, which weakens the binding ability of TKI to EGFR by increasing the affinity of EGFR to ATP [9,10]. Other mechanisms of resistance mainly include activating key downstream effector molecules for the growth and survival of tumor cells identical to EGFR, also called “bypass” resistance mechanisms, such as amplification of the ERBB2 gene encoding the HER2 protein and the expansion of the MET gene encoding the MET tyrosine kinase receptor [11]. BRAF, PIK3CA, KRAS mutations, PTEN deletions, NF-1 deletions, and CRKL amplification have also been reported to mediate TKI resistance through “bypass” resistance mechanisms [12,13]. Therefore, to improve the prognosis of patients with EGFR-mutation NSCLC, the development of novel treatments that offer the possibility to inhibit tumor growth and invasion is particularly important.

Fasudil, which is a selective inhibitor of Rho kinase (ROCK), has been approved by the Japanese and Chinese governments for the clinical treatment of cerebral vasospasm [14]. Further, it has also been used in the treatment of acute lung injury [15], acute ischemic stroke [16], and atherosclerosis. Interestingly, several in vivo and in vitro studies have demonstrated that Fasudil has anti-tumor effects on its own or in combination with other anti-tumor drugs [17,18]. For example, in malignant gliomas, Fasudil has an anti-angiogenic effect and has been shown to inhibit tumor progression in mouse glioma models; in lung cancer, Fasudil can promote the differentiation and maturity of small-cell lung cancer and inhibit tumor growth, and when combined with BI-2536, it leads to synergistic therapeutic effects in KRAS-mutant lung cancers [19]; and by inhibiting ROCK activity, Fasudil can sensitize gemcitabine therapy in pancreatic cancer stem cells [20]. Using pancreatic cancer mouse models, it has also been demonstrated that pretreating pancreatic tumors with Fasudil relaxes the surrounding tissues and enhances the sensitivity of chemotherapeutic drugs [21]. Furthermore, Fasudil has also shown the ability to reverse drug resistance in temozolomide-resistant gliomas by inhibiting the ROCK2/ABCG2 signaling pathway [22]. Additionally, in hematological malignancies, the ROCK-regulated actomyosin cytoskeleton contributes to the survival and growth of TK-driven malignancies; thus, ROCK inhibitors can be used as effective therapeutics either by themselves or in combination with other TKIs for the treatment of oncogenic TK-driven malignancies [23]. These studies have shown that Fasudil has an anti-tumor effect not only when used alone but also in combination with chemotherapy and targeted drugs, increasing the sensitivity of the drugs and improving the therapeutic effect on drug-resistant tumors. However, whether it can be used as an adjunct to EGFR-mutation NSCLC, especially the gefitinib-resistant EGFR-mutation NSCLC, remains unknown.

Therefore, in this study, our objective was to investigate the effects of Fasudil on the proliferation and apoptosis of TKI-sensitive mutations and TKI-resistant mutant NSCLC cells. We also evaluated the anti-tumor effects of a novel treatment strategy involving the co-administration of Fasudil and an EGFR-TKI (gefitinib) in gefitinib-resistant EGFR-mutation NSCLC using in vivo and vitro models. The results can serve as a theoretical basis for combination therapy using Fasudil and TKIs in future clinical trials.

## 2. Methods

### 2.1. Cell Lines and Cell Culture

Human normal lung epithelial cell line HBE (ATCC CRL-2078), human NSCLC cell line NCI-H1975 (EGFRL858R+T790M, gefitinib-resistant, ATCC CRL-5908), and HCC827 (ATCC CRL-2868) were purchased from the American Type Culture Collection (ATCC, Manassas, VA, USA), while human NSCLC cell line PC9 (EGFRexon19 del; gefitinib-sensitive) was kindly provided by Shenzhen People’s Hospital (Shenzhen, China). Gefitinib-resistant HCC827 (HCC827GR) cells were established at increasing concentrations of gefitinib for approximately five months. These cell lines were cultured at 37 °C in RPMI-1640 medium supplemented with 10% fetal bovine serum (FBS) and 1% penicillin/streptomycin at 37 °C in a humidified atmosphere of 5% CO2. Fasudil (HA-1077) and gefitinib (ZD1839) were purchased from MedChem Express (Monmouth Junction, NJ, USA). FBS, antibiotics, and RPMI-1640 medium were purchased from Gibco (Carlsbad, CA, USA).

### 2.2. CCK-8 Assay

Cell survival rates were estimated via CCK-8 assay. Specifically, ~5000 cells were seeded in 96-well plates containing 100 μL of the culture medium in each well. After culturing with different doses of Fasudil for 24, 48, and 72 h, each well was again incubated with 100 μL of the medium containing 10 μL CCK-8 solution for 1 h. Thereafter, absorbance measurements were performed at 450 nm.

### 2.3. Colony Formation Assay

The cells were seeded in six-well culture dishes at a density of 1000 cells per well. After culturing for 14 days in different doses of Fasudil, the cells were fixed and stained with 0.5% crystal violet to visualize the cell colonies. The experiments were performed in triplicates.

### 2.4. Apoptosis Analysis

After incubation with different concentrations of Fasudil for 24 h, 48 h, or 72 h, the cells were harvested via trypsinization without EDTA. Thereafter, they were resuspended in 200 μL of the binding buffer and stained with 2 μL Annexin V-PE and 3 μL 7-AAD (BD Biosciences, San Jose, CA, USA) at room temperature in the dark for 15 min. This was followed by flow cytometry (FACS LSRII, BD Bioscience, San Jose, CA, USA), through which cell apoptosis was determined.

### 2.5. Cell Cycle Arrest Analysis

After incubation with Fasudil for 48 h, the cells were collected and fixed in chilled 75% ethanol at −20 °C overnight or longer. Thereafter, the cells were washed twice with PBS to remove residual alcohol followed by resuspension in 250 μL DNA staining solution (PI/RNase Staining Buffer, Cat. No. 550825) at room temperature in the dark for 30 min. Flow cytometry (FACS LSRII, BD Bioscience) was then employed to measure cell cycle arrest, and the results were analyzed using ModFit LT (Verity Software House, Topsham, ME, USA).

### 2.6. Analysis of Drug Interactions

To quantify drug interactions between Fasudil and gefitinib, SynergyFinder was used. All simulations were performed assuming that the two drugs were combined in a non-fixed ratio of doses with variable concentrations of Drug 1 and Drug 2. SynergyFinder could estimate the synergy scores of the proposed drug pairs by three different reference models: Bliss, highest single agent (HSA), and zero interaction potency (ZIP). Synergy scores <−10 represent that the interaction between two drugs is likely to be antagonistic; scores from −10 to 10 mean that the interaction between two drugs is likely to be additive; and scores >10 indicate synergistic interaction between two drugs [24]. Combination Sensitivity Score (CSS) is a visual index, which could be recognized as the inhibition rate at the respective IC50 concentrations of two drugs [25].

### 2.7. RNA-seq

The NCI-H1975 cells were treated with Fasudil (50 and 75 μM) for 24 h. The H1975 cells that did not show any stimulation after saline treatment served as the control group. Total RNA was extracted using TRIzol reagent following the manufacturer’s protocols. The quality of the experimental samples was ensured by testing the concentration, purity, and integrity of the total RNA extracted from the samples. Specifically, after the quality of the samples was confirmed, mRNA purification and chain-specific library preparation were performed. Oligo (dT)-attached magnetic beads were used for mRNA purification, followed by mRNA fragmentation into small pieces using a fragment buffer at the appropriate temperature (94 °C). Thereafter, first-strand cDNA was generated using random hexamer-primed reverse transcription, followed by a second-strand cDNA synthesis. To end repair via incubation, A-Tailing Mix and RNA Index Adapters were added. The cDNA fragments thus obtained were then amplified via PCR, and the resulting products were purified using Ampure XP Beads (Beckman Coulter, High Wycombe, UK), followed by dissolution in EB solution. The quality of the product was then validated using the Agilent Technologies 2100 bioanalyzer (Santa Clara, CA, USA). Further, the double-stranded PCR products from the previous step were heat-denatured and circularized using the splint oligo sequence to obtain the final library. The single-strand circle DNA (ssCir DNA) was formatted as the final library, which was then amplified with phi29 to obtain the DNA nanoball (DNB), with over 300 copies of each molecule. Furthermore, the DNBs were loaded into the patterned nanoarray, and single-end 50-base reads were generated on the BGIseq500 platform (BGI-Shenzhen, Shenzhen, China).

The sequencing data was filtered using SOAPnuke software, BGI Company, Shen Zhen, China (v1.5.2) [26] by: (1) removing reads containing sequencing adapters; (2) removing reads with low-quality base ratios (base quality ≤5), i.e., above 20%; and (3) removing reads with unknown base (‘N’ base) ratio above 5%. Thereafter, the obtained clean reads were stored in FASTQ format. This was followed by mapping to the reference genome using HISAT2, BGI Company, Shen Zhen, China (v2.0.4) [27]. Further, Bowtie2, BGI Company, Shen Zhen, China (v2.2.5) [28] was applied to align the clean reads to the reference coding gene set. This was followed by the determination of gene expression levels using RSEM, BGI Company, Shen Zhen, China (v1.2.12) [29]. The heatmap corresponding to the genes was generated using pheatmap, BGI Company, Shen Zhen, China (v1.0.8) according to the gene expression levels in the different samples. Essentially, differential gene expression (DEG) analysis was performed using DESeq2, BGI Company, Shen Zhen, China (v1.4.5) [30] with Q value ≤ 0.05. To clarify changes in phenotype, GO (http://www.geneontology.org/, accessed on 21 March 2022) and KEGG (https://www.kegg.jp/, accessed on 21 March 2022) enrichment analyses for the annotation of DEGs were performed using Phyper (https://en.wikipedia.org/wiki/Hypergeometric_distribution, accessed on 21 March 2022) based on hypergeometric testing. The significance levels of the terms and pathways were corrected using the Q value based on a rigorous threshold (Q value ≤ 0.05) based on the Bonferroni method [30].

### 2.8. Western Blotting Analysis

The cells were lysed with RIPA buffer (Beyotime, Shanghai, China) containing 1X PMSF and a protease inhibitor cocktail. After centrifugation at 4 °C and 12,000× *g* for 15 min, the concentration of proteins in the supernatant was determined using a Piece BCA protein assay kit (Thermo Fisher Scientific Inc., Waltham, MA, USA). The proteins were separated using 10% sodium dodecyl-sulfate-polyacrylamide gels. Thereafter, they were transferred onto a PVDF membrane using a transfer apparatus, and after blocking with 5% skimmed milk, the membrane was incubated with the primary antibody overnight at 4 °C. The primary antibodies, rabbit anti-EGFR, rabbit anti-phospho-EGFR, rabbit anti-FASN, rabbit anti-LIPIN1, rabbit anti-Insig1, rabbit anti-SCD1, rabbit anti-SREBP1, rabbit anti-pAMPK, rabbit anti-p-AKT, rabbit anti-p-PI3K, and mouse anti-GAPDH, were purchased from Cell Signaling Technology (Danvers, MA, USA). Further, rabbit anti-Srebf1 was purchased from Abcam (Cambridge, MA, USA), while goat anti-rabbit IgG and goat anti-mouse secondary antibodies were purchased from Proteintech (Rosemont, IL, USA).

### 2.9. Real-Time Reverse Transcription PCR (RT-PCR)

Total RNA was isolated using TRizol reagent (Takara, Kusatsu, Japan) according to the manufacturer’s instructions. To synthesize cDNA, a high-capacity cDNA reverse transcription kit (Takara, Kusatsu, Japan) was used. The synthesized cDNA was then employed in quantitative PCR using the SYBR Green ER kit (Takara) according to the manufacturer’s instructions.

### 2.10. Quantification of Neutral Lipid

The lipophilic fluorescence dye BODIPY 493/503 (Invitrogen, Waltham, MA, USA) was used to monitor the content of neutral lipids in NSCLC cells. After being fixed in 4% PFA for 20 min, cells were incubated with BODIPY 493/503 (D3299, Thermo Fisher, Waltham, MA, USA) and DAPI in PBS at RT for 15 min. Finally, the cells were visualized with a fluorescence microscope (Olympus, Tokyo, Japan). A representative image is shown from three independent experiments.

### 2.11. Oleic Acid Rescue

The H1975 and HCC827GR cell lines were treated with ddH_2_O, oleic acid (2%), Fasudil (50 μM), gefitinib (5 μM), Fasudil + gefitinib, and Fasudil + gefitinib + oleic acid, respectively, for 48 h, followed by apoptosis assay.

### 2.12. Nude Mice Xenograft Model

To establish the xenograft tumor model, H1975-mCherry cells (1 × 10^7^ cells in 100 µL PBS and Matrigel, 1:1) were subcutaneously injected into the right armpit of the forelimb of 5–6-week-old male BALB/c nude mice. When the tumor volume reached an average of approximately 50–100 mm^3^, the mice were randomly divided into four groups (*n* = 4) and were treated as follows: Group 1, control (normal saline); Group 2, Fasudil (50 mg/kg); Group 3, gefitinib (5 mg/kg); Group 4, Fasudil (50 mg/kg) and gefitinib (5 mg/kg). Fasudil was diluted in saline, while gefitinib was dissolved in DMSO; both drugs were administered intraperitoneally (i.p.). Further, all the treatments were administered five times per week for 3 weeks consecutively. The sizes of the tumors and the body weights of the mice were determined every other day. Specifically, the tumor sizes were determined by measuring the length (a) and width (b) of the tumors using calipers. Thereafter, tumor volume (V) was calculated using the formula: V (mm^3^) = 1/2*b^2^. At week 3, after the last drug treatment, immunofluorescence images from the mice were acquired using an in vivo IVIS spectrum imaging system (PerkinElmer, Waltham, MA, USA). The fluorescence density was quantified using Living Image software (PerkinElmer). Finally, the mice were sacrificed, and thereafter, tumor samples were collected for weight and volume measurements and IHC analyses. All the animal experiments were performed in accordance with the animal protocols approved by the Institutional Animal Use and Care Committee of Tongji Medical College, Huazhong University of Science and Technology.

### 2.13. Statistical Analysis

All the statistical analyses were performed using GraphPad Prism v7.0 (GRAPH PAD Software Inc., Los Angeles, CA, USA), and the results are presented as the mean ± SD of at least three independent experiments. To compare means, the unpaired Student’s *t*-test and ANOVA were employed, and the levels of statistical significance were set at * *p* < 0.05, ** *p* < 0.01, and *** *p* < 0.001.

## 3. Results

### 3.1. Fasudil has Stronger Growth Inhibition and Pro-Apoptotic Effects on TKI-Resistant-Mutations NSCLC Cells, Compared with TKI-Sensitive-Mutations NSCLC Cells

To investigate the inhibitory effect of Fasudil in TKI-sensitive-mutations and TKI-resistant mutant NSCLC cells, H1975, HCC827, and PC9 cells were treated with different concentrations of Fasudil (25, 50, 75, and 100 μM) for 24, 48, and 72 h. Thus, it was observed that Fasudil treatment inhibited cell proliferation in a dose-dependent manner in these three cell lines. Specifically, Fasudil showed a stronger anti-proliferative effect in gefitinib-resistant H1975 cells than in gefitinib-sensitive cells (HCC827 and PC9). The IC_50_ values corresponding to the indicated treatment times are shown in Figure 1A. Further, to confirm these results, we performed colony formation to clarify the anti-proliferation effects of Fasudil in HCC827, PC9, and H1975 cells. As expected, the number of colonies corresponding to HCC827, PC9, and H1975 cells decreased sharply in the presence of 25 μM Fasudil compared with the control. Furthermore, the number of colonies corresponding to H1975 cells was lower than those corresponding to HCC827 and PC9 cells (Figure 1B,C). To further confirm whether Fasudil inhibited cell proliferation by inducing cell cycle arrest, flow cytometry was performed. Thus, it was observed that the percentage of cells in the G2/M phase increased while that of cells in the G0/G1 and S phases decreased when HCC827, PC9, and H1975 cells were treated with 50 µM Fasudil for 48 h. These results suggest that Fasudil inhibits EGFR-activated-mutation NSCLC cell line proliferation, which may be associated with G2/M cell cycle blocking (Figure 1D). To determine the effect of Fasudil on apoptosis and cell death, an Annexin V–7-ADD dual staining assay was performed via flow cytometry. The apoptotic rates corresponding to HCC827 cells treated with increasing Fasudil concentrations (0, 50, and 75 μM) for 48 h were 3.97, 11.83, and 23.1%, respectively. Those corresponding to the PC-9 cells were 3.43, 20.4, and 32.87%, and those corresponding to H1975 cells were 5.10, 14.54, and 38.4%, respectively (Figure 1E,F). Furthermore, 24 h and 72 h stimulation of Fasudil on aforementioned NSCLC cell lines were performed as well. As illustrated in Appendix A, the mean proportions of apoptotic HCC827 cells treated with 0, 50, and 75 μM Fasudil for 24 h were 8.64, 15.28, and 19.8 and for 72 h were 8.17, 17.51, and 23.16, respectively. With respect to PC9, the mean rates of apoptotic cells stimulated with corresponding Fasudil concentrations (0, 50, and 75 μM) for 24 h reached 3.87, 6.97, and 11.37 and for 72 h rose to 8.86, 21.64, and 32.13, respectively. In regard to gefitinib-resistant H1975 cells, after undergoing 24 h increasing Fasudil stimulation, there were 8.76%, 11.29% and 18.87% apoptotic H1975 cells. As expected, when H1975 cells were treated with Fasudil in various concentrations (0, 50, and 75 μM), the apoptotic rates rapidly rose to 5.55%, 32.3%, and 44.40% respectively.

These observations indicated that Fasudil induced apoptosis in EGFR-mutation NSCLC cell lines in a dose-dependent manner, and 75 μM Fasudil has a stronger apoptosis-promoting effect on gefitinib-resistant H1975 cells than gefitinib-sensitive cells (HCC827 and PC9).

### 3.2. Fasudil Potentiates the Growth Inhibition and Apoptosis Effect of Gefitinib on Gefitinib-Resistant NSCLC Cells

Based on the aforementioned findings, we wondered whether Fasudil would synergize with gefitinib to potentiate the anticancer effect of gefitinib in EGFR-TKIs-resistant NSCLC cells. Firstly, we constructed an HCC827 gefitinib-resistant cell line (HCC827GR) by exposing HCC827 cells to increasing concentrations of gefitinib for approximately five months. The CCK-8 experiment detected the IC_50_ of gefitinib, revealing that the IC_50_ of HCC827 was 8.34 μM, the IC_50_ of HCC827GR was 46.42 μM, the resistance index of HCC827GR was 5.57, and the IC_50_ of H1975 was 27.94 μM. The IC_50_ of PC9 was 2.68 mΜ (Figure 2A). We then used CCK-8 experiments to detect the effect of Fasudil in collaboration with gefitinib on the inhibition of gefitinib-resistant cell proliferation (Figure 2B). The results reveal that the indicated concentration of Fasudil with different concentrations of gefitinib prominently increased gefitinib cytotoxicity and reduced the IC_50_ against gefitinib. Subsequently, the gefitinib-resistant cell lines HCC827GR and H1975 were exposed to vehicle (DMSO, NT), gefitinib (15 μM), Fasudil (75 μM), or gefitinib (15 μM) +Fasudil (75 μM) for 48 h, followed by Annexin V staining. The results obtained indicate that gefitinib in combination with Fasudil showed a significantly stronger apoptosis effect in the gefitinib-resistant cells than Fasudil- or gefitinib-only treatments (Figure 2C,D).

To confirm that the interaction between Fasudil and gefitinib is not simply additive, we simulated the interaction of the two drugs in SynergyFinder Web [31]. The mean ZIP score (a typical model for calculating synergy score) of H1975 cells treated with Fasudil in combination with gefitinib was over 10, which indicated synergistic effect of two drugs in the whole range of concentrations we set, and the mean ZIP score of HCC827GR cells was 6.91, as the two drugs appeared to have a slight antagonistic effect when Fasudil was around 75 μM and gefitinib was in 8~32 μM (Figure 2E). In addition to ZIP score, both HSA and Bliss score supported synergistic effect (Figure 2F). Recently, CSS score was recommended to assess drug co-action level [25]. As Figure 2F shows, the CSS scores of H1975 and HCC827GR were 80.75 and 74.72, respectively, both of which were over 50. Overall, Fasudil and gefitinib had synergistic effect on H1975 and HCC827GR cells. As a result, Fasudil re-sensitizes gefitinib-resistant cells to gefitinib.

To further clarify the anti-tumor effects of Fasudil in combination with gefitinib, we examined the effects of this drug combination on EGFR/PI3K/AKT signaling pathways in HCC827GR and H1975 cells (Figure 2G). In both cell lines, gefitinib had a partial inhibitory effect on the activation of PI3K/AKT/mTOR, which act downstream of EGFR; meanwhile, Fasudil hardly had any effect. However, co-administration of Fasudil with gefitinib caused an apparent reduction in p-PI3K, p-AKT, and p-mTOR levels. Collectively, given the limited effect of Fasudil or gefitinib alone in TKI-resistant cell lines, these data suggest that combined treatment with Fasudil and gefitinib can synergistically enhance the inhibition effect of gefitinib and revert resistance to gefitinib.

### 3.3. Fasudil Perturbs the Gene Profile Associated with Lipid Metabolism in Gefitinib-Resistant NSCLC Cells

To further clarify the mechanisms by which Fasudil inhibits the survival and expansion of EGFR-resistant-mutations NSCLC cells, we performed RNA-seq analyses involving H1975 cells to identify subsets of sensitive genes resulting from Fasudil treatment (the DEGs were defined as log expression fold changes, |logFC| > 1, with *p* < 0.05). Interestingly, a comparison of the NC group and group A (Fasudil 50 μM) showed that 330 and 709 genes were upregulated and downregulated, respectively, while a comparison of the NC group and group B (Fasudil 75 μM) showed that 571 and 1043 genes were upregulated and downregulated, respectively (Figure 3A). In general, the volcano plot showed more DEGs in the B group (Fasudil 75 μM) than in the A group (Fasudil 50 μM), and the number of downregulated genes was greater than the number of upregulated genes. Further, we extracted the genes that were downregulated with corresponding FC > 2 and further analyzed them via pathway-enrichment analyses using KEGG and GO databases. The results of the KEGG pathway-enrichment analysis indicate that Fasudil treatment downregulated genes that were significantly associated with pathways related to steroid biosynthesis, metabolic pathways, and the IL-17 signaling pathway (Figure 3B). Furthermore, the most enriched DEGs in terms of GO were the sterol biosynthetic process (GO:0016126), lipid metabolic process (GO:0006629), immune system process (GO:0002376), regulation of cholesterol biosynthetic process (GO:0045540), and cholesterol biosynthetic process (GO:0006695) (Figure 3C). Additionally, as shown in the heatmap in Figure 3D, the expression of genes associated with lipid metabolism pathways, such as FASN, SCD1, SREBF1, SREBF2, ACSL1, ACLY, INSIG1, and FADS1, was significantly downregulated. These results suggest that Fasudil may inhibit cell proliferation and survival by regulating lipid metabolism in drug-resistant cells. Seven DEGs, *ACAT2*, *ACLY*, *FASN*, *INSIG1*, *SCD1*, and *LIPIN1*, were selected to verify the reliability of the microarray profiling data using RT-qPCR. Consistent with the microarray results, the validation analyses indicate that the expression levels of the seven abovementioned genes in H1975 cells decreased significantly following Fasudil treatment compared with those in the untreated cells (Figure 3E). Next, to further verify the effects of Fasudil on lipid metabolism in gefitinib-resistant NSCLC cells, we detected the protein expression of lipid-metabolism-associated molecules in gefitinib-resistant NSCLC cells after Fasudil treatment. Our results show that Fasudil effectively reduced the protein expression levels of FASN, SCD1, and SREBP1 in both H1975 and HCC827GR cells (Figure 3F). Furthermore, we evaluated the effect of Fasudil on intracellular lipids levels by lipophilic dye BODIPY 493/503. After Fasudil treatment, the extent of BODIPY staining was significantly reduced in both H1975 and HCC827GR cells. (Figure 3G). Taken together, these results indicate that Fasudil could decrease intracellular lipids levels of gefitinib-resistant NSCLC cells.

### 3.4. Combined Treatment with Fasudil and Gefitinib Decreases Intracellular Lipid Accumulation in Gefitinib-Resistant NSCLC Cells

Previous studies have shown that abnormal accumulation of intracellular lipids was responsible for gefitinib resistance and inhibition of intracellular lipid synthesis can reverse gefitinib resistance [30,31,32,33]. Our results show that Fasudil can reduce intracellular lipids levels and increase the sensitivity of gefitinib in gefitinib-resistant NSCLC cells. To further investigate whether the inhibitory effects of Fasudil combined with gefitinib in gefitinib-resistant NSCLC cell lines was mediated by decreased intracellular lipid accumulation, we detected the lipid levels of resistant cells when gefitinib is used alone or in combination with Fasudil. The results show that the intracellular lipid levels of HCC827/GR and H1975 cell lines were increased when treated with gefitinib alone, while Fasudil alone or combination with gefitinib can significantly decrease intracellular lipid levels (Figure 4A). Given the fact that the lipid droplets in NSCLC undergo significant changes before and after gefitinib treatment, we further detected the expression of key molecules involved in lipogenic functions in gefitinib-resistant cells after treatment with Fasudil/gefitinib alone or in combination. Our results show that the protein expression levels of FASN, SCD1, and SREBP1 were further decreased in H1975 and HCC827GR cells after combination therapy compared with Fasudil monotherapy (Figure 4B). The results show that gefitinib could enhance the lipogenic in NSCLC cell lines with gefitinib resistance and Fasudil could reverse this effect.

To further confirm that Fasudil re-sensitizes resistant cells to gefitinib through decreasing intracellular lipid level, we conducted an oleic acid rescue experiment on H1975 and HCC827GR treated with Fasudil and gefitinib. The mean apoptosis rates of H1975 and HCC827GR cells in oleic acid rescue group significantly decreased compared with the Fasudil combination with gefitinib group, with rates of 10.82% vs. 32.43% and 14.95% vs. 26.62%, respectively, and oleic acid itself did not induce significant apoptosis (Appendix A). These results strongly support the importance of lipid metabolism for Fasudil to re-sensitize resistant cells to gefitinib.

### 3.5. Fasudil Reduces the Expression of Molecules Related to Fatty Acids Synthesis Via Activating AMPK

AMP-activated kinase (AMPK) is widely known to be a key molecule in metabolic conditions. Previous reports showed that AMPK acts as a downstream regulator of ROCK1 in the context of hepatic lipid homeostasis [32]. To investigate the inhibitory effect of Fasudil on intracellular lipid accumulation, the levels of ROCK1 and AMPK phosphorylation were evaluated by western blotting. As shown in Figure 5A, phosphorylation of AMPK was significantly increased by the Fasudil treatment, and ROCK1 was significantly reduced in a concentration-dependent manner. Next, we attempted to inhibit the phosphorylations of AMPK by pretreatment of H1975 cells with compound C (CC), an AMPK inhibitor. As shown in Figure 5B, Fasudil-induced downregulation in SCD1 protein expressions was abolished by treatment with CC. These results indicate that Fasudil regulated lipogenic gene expressions via AMPK signal pathway.

### 3.6. Fasudil-Only Treatment or Fasudil Treatment in Combination with Gefitinib Mitigates the Growth of EGFR-Mutation NSCLC In Vivo

Next, we evaluated the efficacy of Fasudil monotherapy or in combination with gefitinib with respect to the mitigation of EGFR-mutated NSCLC cell proliferation using H1975-cell-line-derived xenograft animal models. The mice were treated with vehicle, Fasudil (50 mg/kg), gefitinib (5 mg/kg), or both for 3 wks. The representative tumors from each group are shown in Figure 6A. Further, the results of this animal experiment show significantly suppressed tumor growth and fluorescence signals following the combination treatment compared with monotherapy (Figure 6B, C). Additionally, the tumor volumes corresponding to the vehicle, Fasudil, gefitinib, and Fasudil + gefitinib treatments were 871 ± 132.67, 591.9 ± 69.09 (** *p* < 0.01), 627 ± 43.77 (** *p* < 0.01), and 434.4 ± 73.62 mm^3^ (** *p* < 0.01), respectively (Figure 6D). The tumor inhibition rate corresponding to the combination treatment was also found to be 50% higher than that corresponding to the vehicle group, and after weighing the extracted tumor tissues, the group treated with Fasudil showed reduced tumor weight compared to the vehicle group (*p* < 0.05). Moreover, the combination treatment group (Fasudil and gefitinib) showed a lower tumor weight than that of the monotherapy group (*p* < 0.05, Figure 6E). However, there were no significant differences between the body weights of the control and treated mice (Figure 6F), and histological analysis showed no significant organ histological changes after drug treatment, indicating that Fasudil monotherapy as well as the Fasudil treatment in combination with gefitinib were well-tolerated (Figure 6G). Tumor masses were isolated and analyzed by HE and IHC. HE staining of the tumor tissue revealed that the combined administration group exhibited a larger death area. Simultaneously, the proliferation of tumor cells in vivo was detected by ki-67 staining. As shown in Figure 6H, the number of ki-67 positive cells was lower in the combination group than in the other groups.

## 4. Discussion

Despite the significant advances in EGFR-mutation NSCLC treatment, current therapies are still ineffective in many patients owing to late-stage diagnosis and the acquisition of resistance to EGFR TKIs [33]. Fasudil, which is the most commonly used ROCK inhibitor, has been studied in many fields including cardiovascular disease, inflammation, and cancer research [34]. Fasudil has been demonstrated to have anti-tumor effects through inhibiting tumor growth and invasiveness in several cancers [18]. In addition, some studies have revealed that Fasudil could suppresses the NSCLC invasive phenotype and promotes small-cell lung cancer maturation and apoptosis [35]. When combined with BI-2536 (an inhibitor of polo-like kinase 1), Fasudil suppressed KRAS-mutant lung cancer growth in a corresponding LSL-KRAS (G12D) mouse model and in a patient tumor explant mouse model of KRAS-mutant lung cancer [19]. An early study described that Fasudil inhibited A549 cell proliferation through downregulating VEGF expression [36]. However, the anti-tumor role of Fasudil in EGFR-mutation NSCLC as well as its mechanism are largely unknown.

In this study, we first investigated the inhibitory effect of Fasudil on the proliferation and apoptosis of EGFR-mutation NSCLC cells. Our results show that Fasudil treatment inhibited cell proliferation and clone formation and also induced apoptosis and G2/M-phase arrest in the EGFR-mutation NSCLC cells. They also show that its inhibitory effect in gefitinib-resistant H1975 cells was stronger than that in gefitinib-sensitive cells. In addition, our cell viability assay confirmed that the antiproliferative effects of gefitinib on gefitinib-resistant cell lines were enhanced by Fasudil treatment in a dose-dependent manner. In the apoptosis assay, we discovered that Fasudil significantly enhances gefitinib-induced apoptosis in gefitinib-resistant cell lines. Our results indicate that gefitinib combined with Fasudil exhibited synergized inhibitory effects in gefitinib-resistant cells (HCC827GR and H1975). Notably, the results of our animal experiment also show that this combination strategy has anti-tumor potential in H1975 mouse models in vivo. Further, functional enrichment analysis of downregulated genes in Fasudil-treated H1975 cells suggested that Fasudil treatment influences cell metabolic processes, especially lipid metabolism.

Tumor metabolism reprogramming has been extensively studied in recent years [37]. Proliferating cancer cells rely mostly one de novo adipogenesis, which is crucial for membrane biosynthesis and signaling molecules. Few studies have focused on the implications of lipid metabolism in lung cancer biology. Previous studies found that NSCLC cell lines with EGFR mutation are more dependent on lipid metabolism than EGFR wild-type NSCLC cell lines [38]. Chen demonstrated that lovastatin, a cholesterol suppressor, can restore sensitivity of gefitinib in gefitinib-resistant cell lines by inhibiting the phosphorylation levels of EGFR, Akt-1, MEK1/2, and ERK1/2 [39]. It has also been reported that lipid-metabolism-related molecular inhibitors (SCD and SREBP) enhance gefitinib sensitivity by decreasing cell membrane fluidity and EGFR signaling [13,40,41,42]; In our study, we reported that Fasudil can enhance sensitivity of gefitinib in gefitinib-resistant cell lines by decreasing intracellular lipids levels. In fatty liver disease, ROCK activation is necessary to regulate the endocannabinoid-mediated lipogenic program through suppressing AMPK activity [43]. As a ROCK inhibitor, Fasudil may play a pivotal role in regulating lipid metabolic pathways in lung cancer. The present study provides a new perspective on the mechanism of Fasudil in inhibiting cancer cell growth.

Osimertinib inhibited both EGFR-activating and T790M-resistance mutations and was approved for the treatment of NSCLC patients carrying a T790M-resistance mutation [44]. With the widespread use of osimertinib, the incidence of drug resistance would gradually increase [45]. In our study, we observed that Fasudil had stronger inhibitory effect in gefitinib-resistant H1975 cells than that in gefitinib-sensitive cells. Some researches inferred that such resistance mechanisms of osimertinib were similar to those of other first- or second-generation TKIs [46]. Therefore, to enhance the clinical significance of Fasudil here and now, it is worth further studying the synergistic effect of Fasudil and osimertinib, especially in osimertinib-resistant situations.

Fasudil has already been approved by the FDA for clinical use, and Fasudil monotherapy or combined therapy at the doses used in this study was found to be well tolerated; no organ toxicity was observed in any of the treatment groups. Therefore, Fasudil monotherapy, or Fasudil in combination with a TKI, could be a promising therapy strategy for patients who cannot afford expensive targeted therapy drugs or are resistant to TKIs.

This study has some limitations. First, clinical trials were not performed to evaluate the efficacy of the concurrent use of Fasudil and EGFR-TKIs or whether the addition of Fasudil may improve clinical outcomes and delay the occurrence of EGFR-TKI resistance. Second, we did not estimate which type of EGFR mutations were responsible for the sensitivity to Fasudil or how much the mutation type matters.

## 5. Conclusions

Our study demonstrated that Fasudil could effectively inhibit EGFR-mutation cell growth and enhance the sensitivity of gefitinib-resistant NSCLC cells to gefitinib by suppressing intracellular lipid accumulation. Mechanistic investigations showed that Fasudil could reverse gefitinib-induced SCD1 expression by suppressing AMPK activity, and combination therapy had a greater inactivation effect on the EGFR/PI3K/AKT pathway than either treatment alone. These findings highlight the possibility of a combination therapy strategy for NSCLC with EGFR mutation and the inhibition of TKI resistance.

## Figures and Tables

**Figure 1 cancers-14-04709-f001:**
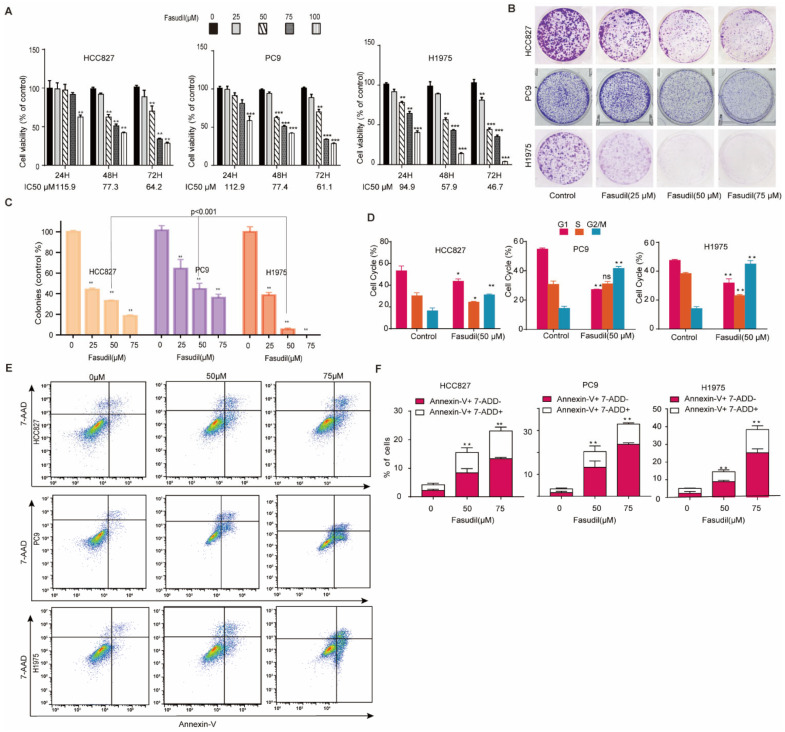
Inhibition of the growth of EGFR-mutation NSCLC cells in vitro by Fasudil. (**A**) Cell viability of EGFR-mutation NSCLC cells (HCC827, PC9, and H1975) treated with Fasudil at the indicated concentrations for 24, 48, and 72 h. The cell viability was analyzed via CCK-8 assay. All values are expressed as the mean ± SD (*n* = 4). (**B**) Cell clone colonies formed by HCC827, PC9, and H1975cells treated with Fasudil (0, 25, 50, and 75 μM). (**C**) Quantification of the colonies in each well. All values are expressed as the mean ± SD (*n* = 3). (**D**) HCC827, PC9, and H1975 cells treated with 50 μM Fasudil for 48 h, stained with PI, and analyzed via flow cytometry. Statistical graph showing the percentages of cells in the G0/G1, S, and G2/M phases in the cell cycle. All values are expressed as the mean ± SD (*n* = 3). (**E**) HCC827, PC9, and H1975 cells treated with Fasudil for 48 h, double-stained with Annexin V-FITC/PI, and analyzed via flow cytometry. (**F**) Quantification of the apoptotic cells. All data are presented as mean ± SD based on triplicate measurements; * *p* < 0.05, ** *p* <0.01 and *** *p* < 0.001 were considered statistically significant.

**Figure 2 cancers-14-04709-f002:**
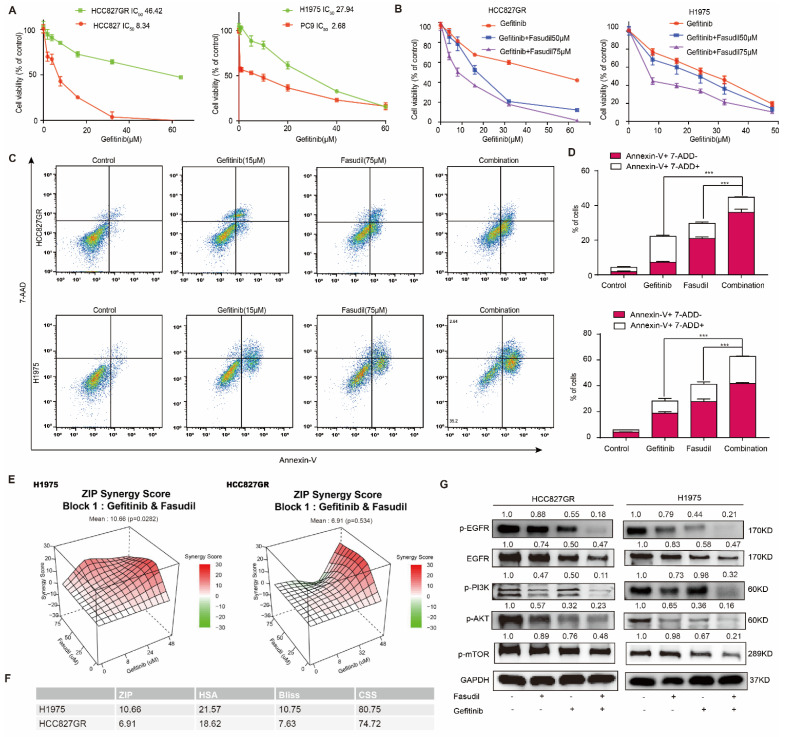
Effects of the combination of gefitinib with Fasudil on the growth inhibition and apoptosis in gefitinib-resistant NSCLC cells. (**A**) HCC827, HCC827GR, PC9, and H1975 cells were treated with various concentrations of gefitinib for 48 h, and the CCK-8 assay was used to determine cytotoxicity. All values are expressed as the mean ± SD (*n* = 4). (**B**) Combined treatment of Fasudil and gefitinib enhances the synergistic cytotoxic effects in gefitinib-resistant NSCLC cells. Gefitinib monotherapy (0, 8, 16, 24, 32, 48 µmol/L) or treatment with gefitinib in combination with Fasudil (50 or 75 µmol/L) for 48 h. (**C**) HCC827GR and H1975 cells subjected to Fasudil and gefitinib monotherapy or Fasudil and gefitinib combination therapy for 48 h. The percentage of apoptotic cells was determined via flow cytometry. (**D**) Quantification of the apoptotic cells, data are presented as mean ± SD based on triplicate measurements. *** *p* < 0.001 were considered statistically significant. (**E**) The 3D plot of ZIP synergy score (left: H1975, right: HCC827GR). The mean ZIP scores of H1975 and HCC827GR were 10.66 and 6.91, respectively, which meant the interaction was likely to be synergistic. (**F**) The ZIP score, HSA score, Bliss score, and CSS score of H1975 and HCC827GR cells. (**G**) Western blotting of p-EGFR, EGFR, p-AKT, and p-PI3K. GAPDH was included as a loading control. Protein expression was quantified by ImageJ software. The uncropped blots are shown in Appendix A.

**Figure 3 cancers-14-04709-f003:**
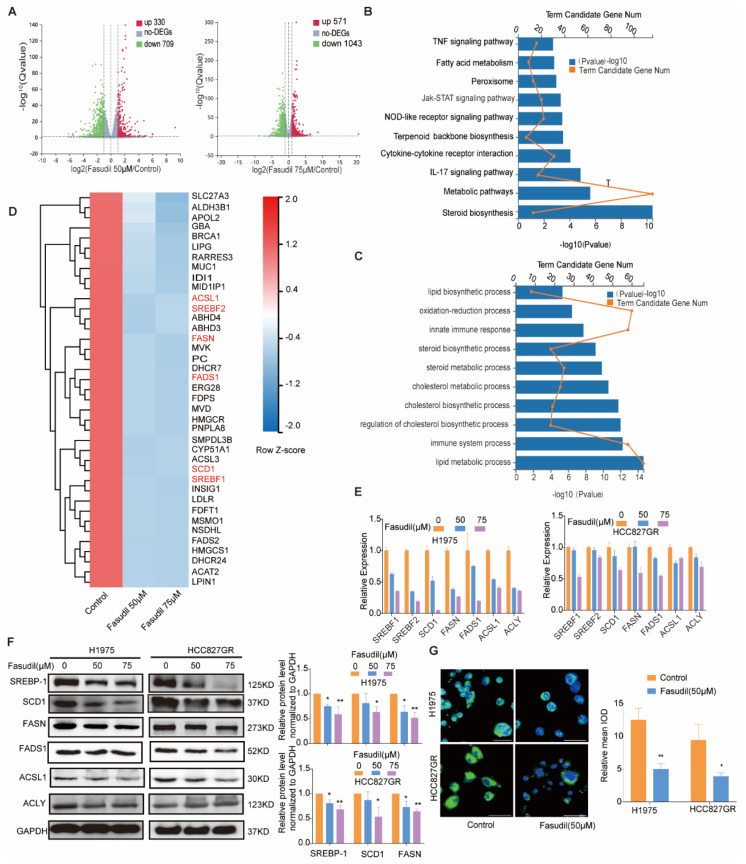
Changes in the gene-expression profile of Fasudil-treated H1975 cells. (**A**) Volcano plot showing the distribution of -log10 (FDR) vs. log2 (Fold Change). The red points indicate upregulation (*p* < 0.05) and green points indicate downregulation (*p* < 0.05), while the gray points indicate P ≥ 0.05. (**B**) KEGG pathway-enrichment analysis for Fasudil-induced downregulated genes. (**C**) GO pathway-enrichment analysis results. (**D**) Heatmap of the expression levels of lipid-metabolism-associated genes (Z scores were used). The red and green areas indicate high and low expression levels, respectively. (**E**) RT-qPCR validation of the downregulation of lipid-metabolism-associated genes (SREBF-1, SREBF-2, SCD1, FASN, FADS1, ACSL1, and ACLY) in H1975 cells after Fasudil treatment. All values are expressed as the mean ± SD (*n* = 3). * *p* < 0.05 and ** *p* < 0.01. (**F**) Western blot analysis of SREBP-1, SCD1, FASN, FADS1, ACSL1, and ACLY expression in HCC827/GR and H1975 cells treated with Fasudil (50, 75 µM) for 48 h. (**G**) H1975 and HCC827/GR cells were stained with BODIPY 493/503 (green) and counterstained with DAPI (blue) after 48 h of exposure to Control (ddH_2_O) or Fasudil (50 μM). Lipid accumulation of indicated cell lines was evaluated by measuring green fluorescence by Image J. The uncropped blots are shown in Appendix A.

**Figure 4 cancers-14-04709-f004:**
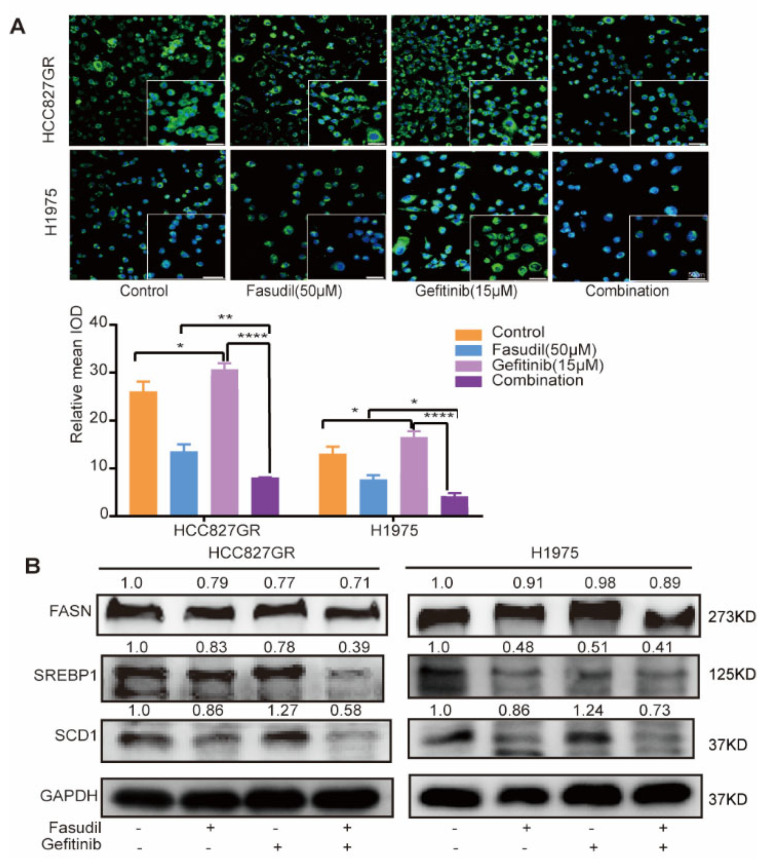
Combined treatment with Fasudil and gefitinib decreases intracellular lipid accumulation in gefitinib-resistant NSCLC cells. (**A**) H1975 and HCC827/GR cells were stained with BODIPY 493/503 (green) and counterstained with DAPI (blue) after 48 h of exposure to control (ddH_2_O), Fasudil (50 μM), gefitinib (15 μM), or Fasudil (50 μM) + gefitinib (15 μM). Lipid accumulation of indicated cell lines was evaluated by measuring green fluorescence by Image J. Scale bar: 50 μm. * *p* < 0.05, ** *p* < 0.01 and **** *p* < 0.0001. (**B**) Western blot analysis of FASN, SREBP-1, and SCD1 expression in HCC827/GR and H1975 cells treated with control (ddH_2_O), Fasudil (50 μM), gefitinib (15 μM), or Fasudil (50 μM) + gefitinib (15 μM) for 48 h. Each gene expression was normalized to glyceraldehyde-3-phosphate dehydrogenase (GAPDH). The uncropped blots are shown in Appendix A.

**Figure 5 cancers-14-04709-f005:**
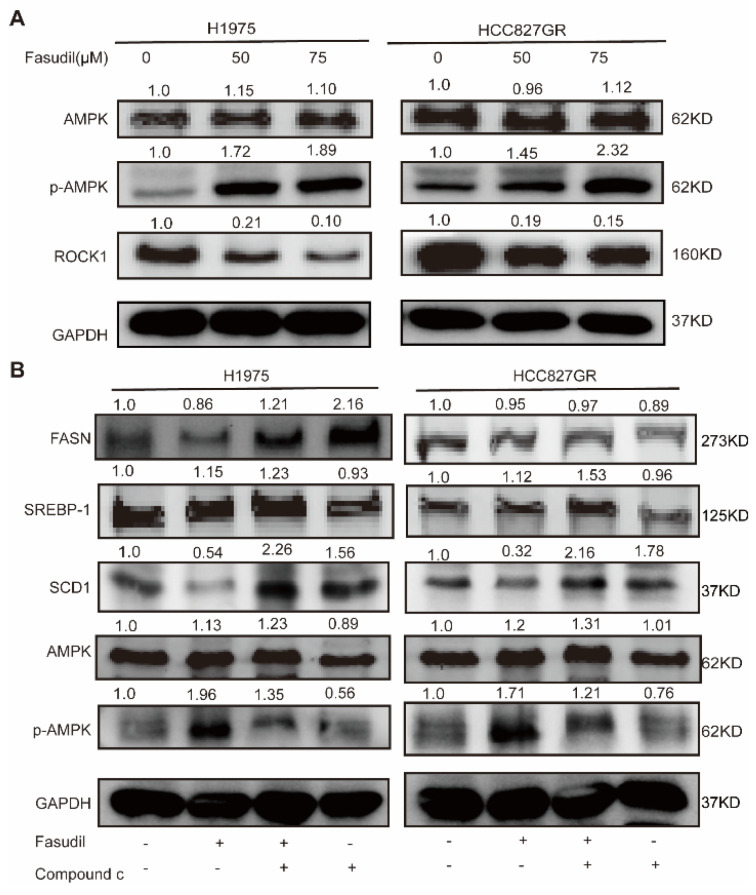
Effects of Fasudil on AMPK phosphorylations in gefitinib-resistant NSCLC cells. (**A**) H1975 and HCC827/GR cells were treated with Fasudil for 48 h. Western blot analysis of phosphorylation of AMPK and ROCK1 expression. (**B**) Phosphorylation of AMPK and FASN, SREBP-1, and SCD1 expression were also examined in the presence of compound C, an AMPK inhibitor. Each gene expression was normalized to glyceraldehyde-3-phosphate dehydrogenase (GAPDH). The uncropped blots are shown in Appendix A.

**Figure 6 cancers-14-04709-f006:**
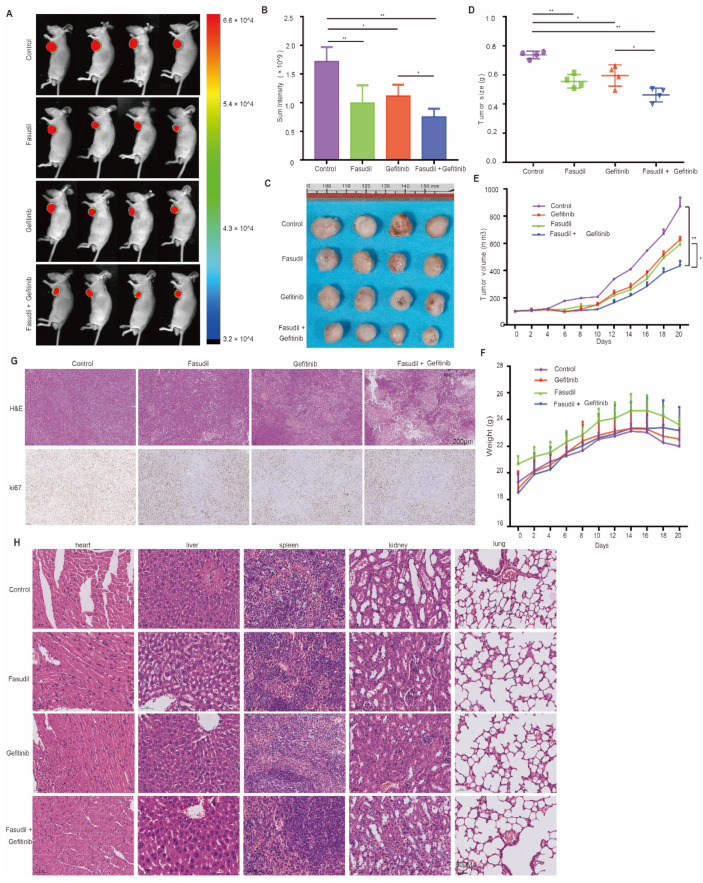
Mitigation of EGFR-mutation NSCLC cell proliferation in vivo by Fasudil monotherapy or Fasudil treatment combined with gefitinib. (**A**) Representative images of subcutaneous tumors resulting from the treatment of the H1975 cell line with control, Fasudil (50 mg/kg), gefitinib (5 mg/kg), or combination treatments. *n* = 4; the fluorescence signal intensity is shown in (**B**–**D**). Tumor volume in H1975 tumorigenic mice following control, Fasudil (100 mg/kg), gefitinib (100 mg/kg), or combination treatments; *N* = 4. (**E**) Quantification of the weight of subcutaneous tumors from H1975 cells treated with control, Fasudil (50 mg/kg), gefitinib (5 mg/kg), or combination treatments; *n* = 4. (**F**) Weights of mice recorded every two days. The data are represented as mean ± SEM. (**G**) Representative images of HE staining of heart, liver, spleen, kidney, and lung. (**H**) Representative images of HE and IHC for ki67 in subcutaneous tumors formed by H1975 cell line treated with control, Fasudil (50 mg/kg), gefitinib (5 mg/kg), or combination. * *p* < 0.05 and ** *p* < 0.01 were considered statistically significant.

## Data Availability

The datasets used and/or analyzed during the current study are available from the corresponding author on reasonable request.

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
