# Peer review of "Fasudil Increased the Sensitivity to Gefitinib in NSCLC by Decreasing Intracellular Lipid Accumulation"

_cancers, 2022, doi:10.3390/cancers14194709_

Round 1
Reviewer 1 Report
The work in this manuscript uses molecular biological and cellular methods to investigate the anti-tumor effects of Fasudil on the EGFR-mutation NSCLC and corresponding molecular mechanism. Thus, they evaluated the inhibitory effects of Fasudil monotherapy and combination therapy on resistant EGFR-mutation NSCLC in vitro and in vivo, respectively. The research is properly executed, and results are well analyzed. The presentation is mostly detailed and will certainly permit replication. However, a part of research is not well designed, and some critical results need to be provided. Because of this, a part of the conclusion is relatively speculative and must be more cautiously formulated.
Major issues
1) One of major conclusion in this manuscript is Fasudil re-sensitize Gefitinib-resistant EGFR-mutation NSCLC by suppressing intracellular lipid accumulation. Indeed, their results showed that Fasudil alone and combination of Fasudil and Gefitinib could decrease intracellular lipid level and molecule level related to fatty acids synthesis. this prove correlation between Fasudil-induced decrease in intracellular lipid level and resensitization to Gefitinib. However, the direct evidence or results is barely missing to prove causation between Fasudil-induced decrease in intracellular lipid level and resensitization to Gefitinib. Perhaps, rescue intracellular lipid level in cells treated with combination of Fasudil and Gefitinib via overexpressing signaling protein in related pathway.
2) Method/Apoptosis analysis: Authors designed 24, 48 and 72 hours of Fasudil treatment for evaluating cell survive rate. Why was apoptosis analysis evaluated only for 48 hours of incubation? Or these data are not shown in this manuscript? I recommend to show these data in maintext or supplementary.
3) Method/RNA-seq: Provide more details for experimental procedure of RNA-seq. For line 146-147, clarify which cell line and treatments was served as the control group. For 153, clarify what the appropriate temperature is.
4) Method/ Nude mice xenograft model: 10% DMSO solution should be set as control, for confirming that 10% DMSO doesn’t have anti-tumor effects on NSCLC.
5) Result 1 : There are inconsistence between results of different assays. HCC827 and pc9 has similar viability under 25, 50, 75 uM Fasudil treatment for 72 hours, while colony number of PC9 decreased sharply under same concentration of Fasudil after 14 days. As shown in Figure 2B, it seems that pc9 and H1975 have similar colony numbers under 25 uM Fasudil incubation (after 14 days), which might attenuate your conclusion “Fasudil has stronger growth inhibition and pro-apoptotic effects on resistant cells than sensitive cells.” If there is significant difference in colony number between PC9 and H1975, p-value should be shown.
6) Result 2: The result is not sufficient to support the conclusion, which makes conclusion somewhat imprecise. Results in this section demonstrate combination of Fasudil and Gefitinib has greater anti-tumor effect on TKI-resistant NSCLC than Fasudil and Gefitinib alone. However, this is not adequate to conclude Fasudil could re-sensitize TKI-resistant NSCLC to Gefitinib, or Fasudil increase gefitinib cytotoxicity. In result 1, Fasudil is proved having anti-tumor effects on TKI-resistant NSCLC. More evidence or results are required to prove that combination of Fasudil and Gefitinib has synergistic antitumor effects, instead the effects of two drugs are simply added. This conclusion must be more cautiously formulated.
Minor issues
1) Line 47: Reference 4 is secondary source and primary source need to be cited, such as PMID: 24631357, PMID: 22785351.
2) Line 51: Reference 5 is incorrectly cited at this point and does not have sentences or data to describe “frequency of EGFR activation mutation in Caucasian NSCLC populations”.
3) Line 54: Reference 6 is incorrectly cited. This data should be from PMID: 20526205.
4) Sentence at line 57-58 needs references to be supported.
5) Line 102, Line 238: “EGFR-sensitive” and “EGFR-resistant” should be “TKI-sensitive” and “TKI-resistant”.
6) Method /cell cycle arrest analysis (line 141): Provide the more detail about DNA staining solution.
7) Figure.1 E: Tick labels is too blur. Please provide high-resolution figures.
8) Figure.2 C: Tick labels is too blur. Please provide high-resolution figures
9) Line 326: “group B (Fasudil 50 uM)” should be 75 uM.
10) Figure.3 F: Protein expression need to be quantified as done for Figure. 2 E.
11) Figure.4 G: Bar size need to be provided.
Reviewer 2 Report
I’m appreciative of your hard work and my privilege to review your article. This article is well written with fairly nice English for great clarification. The authors tried to demonstrate the potential clinical values of Fasudil. The result was thrilling and while further clinical trials are warranted, the authors deserve acknowledgment. However, Osimertinib has been commonly used nowadays to substitute the first or second generation TKI failure due to T790M mutation. One may wonder what role of Fasudil in such clinical scenarios. The third generation of TKI has only been mentioned once in this article, I’d like the authors to address such issue in their discussion. Please elaborate the significance of Fasudil when Osimertinib is now widely prescribed worldwide.
Round 2
Reviewer 1 Report
Authors' responses and revised manuscript solved comprehensively problems I raised in comments. Specifically, the oleic acid rescue data clearly proved causation between Fasudil-induced decrease in intracellular lipid level and resensitization to Gefitinib. And, synergistic analysis provide compelling evidence supporting that combination of Fasudil and Gefitinib is synergistic instead additive.
In addition, more data and details added in maintext and supplementary clarified the unclear and confusing contents in original manuscript. All of suggested minor issues has been revised.